# Position: The Systemic Lack of Agency in Visual Reasoning

**Yizhao Huang** [1 2 *] **Haoyang Chen** [1 2 3 *] **Shiqin Wang** [1 2 *] **Pohsun Huang** [1 2 *] **Jiayuan Li** [3 4] **Haoyuan Du** [1 2]
**Yandong Shi** [1 2] **Zheng Wang** [1 2 3 †] **Zhixiang Wang** [5 †]

## Abstract

This paper argues that a systemic lack of *Agency* constrains the implicit reasoning capabilities of current Vision-Language Models (VLMs). Implicit reasoning refers to the ability to autonomously discover and utilize *hidden* visual evidence to bridge information gaps, rather than merely relying on explicitly specified targets. This capacity underlies human visual understanding and everyday reasoning. We argue that this limitation arises from a tendency to approach visual reasoning primarily as passive semantic retrieval, rather than as active, situated reasoning that depends on autonomous visual exploration. As a result, most existing benchmarks primarily assess *Passive Capacity*, leaving this aspect of reasoning largely unmeasured. To address this gap, we introduce the Visual Implicit Reasoning Diagnosing Benchmark (V-IRD), which targets this missing quadrant by requiring models to derive answers strictly through autonomous visual analysis. Our results show that, despite strong retrieval abilities, prominent VLMs struggle to utilize reference objects and to attend to visual evidence that requires self-directed inquiry. Simply put, strong semantic recognition does not equate to active visual exploration, revealing a critical gap in current VLMs.

---

*Equal Contribution, †Corresponding author. [1]National Engineering Research Center for Multimedia Software, Institute of Artificial Intelligence, School of Computer Science, Wuhan University [2]Hubei Key Laboratory of Multimedia and Network Communication Engineering [3]Zhongguancun Academy, Beijing, China. 100094 [4]School of Automation, Beijing Institute of Technology [5]Shanda AI Research Tokyo. Correspondence to: Zheng Wang <wangzwhu@whu.edu.cn>, Zhixiang Wang <zhixiang.wang@shanda.com>.

*Proceedings of the 43rd International Conference on Machine Learning*, Seoul, South Korea. PMLR 306, 2026. Copyright 2026 by the author(s).

## 1. Introduction

In cognitive science, perception has long been understood as an active process driven by goal-directed information acquisition rather than passive stimulus reception (Gibson, 2014; Ballard, 1991). Does such agency exist in VLMs? Recent advancements in VLMs have demonstrated remarkable proficiency in semantic recognition and explicit instruction-following (Liu et al., 2023; Singh et al., 2025; Wang et al., 2025a; Yang et al., 2025). However, this proficiency heavily relies on explicit guidance, creating a fundamental gap when transitioning to the unconstrained physical world. In natural settings, visual reasoning is predominantly implicit. Human observers do not wait for explicit instructions to check relevant information for reasoning. Instead, they demonstrate agency: proactively shifting attention from salient foregrounds to subtle background cues, and mining unmentioned evidence to construct coherent reasoning chains (Rose et al., 2023; Wang et al., 2025b).

Our position is that current VLMs suffer from a systemic *Visual Implicit Reasoning Deficit*, defined as the inability to autonomously search for and utilize visual cues not explicitly mentioned in the text prompt. While VLMs possess the capacity to perceive details when given specific guidance, they lack the agency to construct physical arguments from raw visual data. Consequently, when a prompt focuses solely on a target object without pointing to background information, the model shows no strong tendency to actively discover critical hidden information beyond the target. Instead of treating the visual context as a landscape of evidence to be explored, the model acts as a passive observer, overlooking critical visual information.

The complexity of real-world visual reasoning rarely presents itself through explicit instructions. In natural settings, the solution depends on implicit visual evidence which consists of critical geometric or physical cues (Chen et al., 2024; Lu et al., 2024) that are necessary for deduction but absent from the user's prompt. For instance, accurately estimating the dimensions of a non-standard bottle requires the autonomous discovery of a background reference, such as a standard ID card (Yang et al., 2024), rather than relying on the semantic label *bottle*. The core challenge lies not in recognizing the object itself, but in the autonomous retrieval

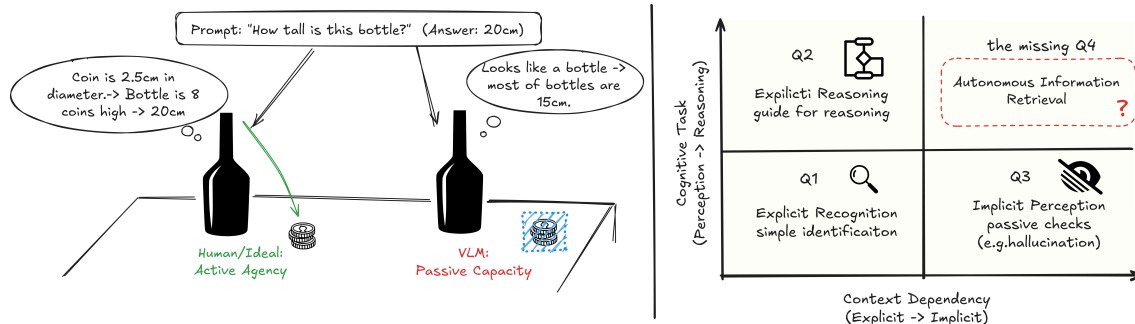

*Figure 1.* **Agency in visual reasoning. Left:** Comparison of active agency *vs.* passive capacity in visual reasoning. Unlike humans actively retrieve implicit visual cues to reason about physical properties, current VLMs tend to ignore the implicit information and give wrong answers. **Right:** Our taxonomy highlights Autonomous Information Retrieval (Q4) as the critical gap. Success here requires visual agency, the ability to actively seek unmentioned visual evidence without explicit prompting which current models lack.

of unmentioned, supportive visual details to build a valid physical argument (Zhao et al., 2025).

In this paper, we formalize this gap as the distinction between visual capacity (what models can see when guided) and visual agency (whether models autonomously seek evidence). Through a series of diagnostic experiments, we provide evidence of this deficit. We demonstrate that without explicit instructions, visual perception and physical reasoning often become decoupled, making it difficult for the model to ground its judgments in implicit visual evidence.

Our contributions are summarized as follows:

- **Concept Definition.** We formalize the *Visual Implicit Reasoning Deficit*, positing that current models fail to autonomously ground physical reasoning in implicit visual cues due to a lack of search agency.

- **V-IRD Benchmark.** To decouple visual discovery from instruction-following, we introduce V-IRD. By enforcing a strict information gap via *Target-Exclusive Prompting*, it compels models to autonomously identify visual cues across four domains, rather than relying on linguistic hints.

- **Evaluation and Analysis.** We propose Threshold Accuracy to separate precise reasoning from estimation. Results reveal a significant performance drop in mainstream VLMs under implicit settings, highlighting a critical deficit in visual agency.

## 2. The Blind Spot in Current Evaluations

Current evaluation methodologies are systematically biased: they measure *visual capacity* while neglecting *visual agency*. We analyze three key areas where this blind spot persists.

### 2.1. Explicit VQA: Externalized Attention

While benchmarks like ColorBench (Liang et al., 2026) and MMIE (Xia et al., 2025) target fine-grained recognition, and V* (Wu & Xie, 2024) introduces guided visual search, they all predominantly operate with explicit instruction.

These benchmarks (Zhang et al., 2025a; Weng et al., 2025; Chia et al., 2024) effectively externalize the visual planning process. The user acts as the attention manager (Hutchins, 1995) by explicitly pointing to the target. High scores in these metrics confirm the model's grounding capability but mask its inability to autonomously identify relevant visual information without direct supervision.

Recent concurrent works like AdaptVision (Lin et al., 2025) and DeepEyes (Zheng et al., 2025) address active visual execution by employing reinforcement learning to dynamically zoom in on local details. However, effective execution requires foundational intent. While these methods provide mechanisms for active perception, our work formalizes its essential prerequisite: the lack of visual agency. Without the intrinsic agency to autonomously seek out unmentioned clues, models struggle to determine where or why to focus, fundamentally bottlenecking the potential of such advanced zooming mechanisms.

### 2.2. Hallucination: Commission vs. Omission

In work on hallucination mitigation, prevailing research such as the advanced diagnostic suite HallusionBench (Guan et al., 2024) and NOPE (Lovenia et al., 2024) predominantly targets errors of commission. These efforts (Li et al., 2023; Rostamkhani et al., 2025; Fu et al., 2024) specifically address the fabrication of non-existent objects or the over-reliance on language priors. Such a perspective largely overlooks implicit neglect which constitutes a critical error of omission.

In this scenario the failure lies not in generating false infor-

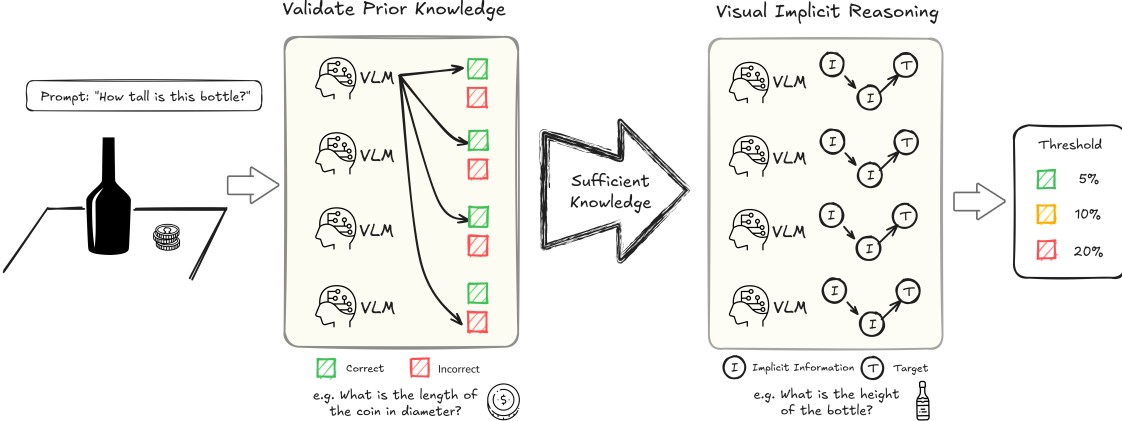

*Figure 2.* **Framework for verifying implicit reasoning capability.** To rigorously assess implicit reasoning independent of knowledge retrieval, we employ a filtered evaluation pipeline. We first validate prior knowledge across domains to ensure the model possesses the necessary factual basis, eliminating errors caused by knowledge gaps. Verified instances then proceed to Visual Implicit Reasoning, where the model needs to autonomously chain Implicit Information to derive the Target answer. This design ensures that performance metrics strictly reflect the model's capability to reason with unstated visual cues rather than its memorized knowledge base.

mation but in failing to utilize existing visual context (Tong et al., 2024; Li et al., 2024c). When a model bypasses the necessary visual search for details like a reference scale to directly generate an answer, it is not hallucinating in the traditional sense but rather suffering from a lack of agency to visually verify its reasoning (Seth et al., 2025).

### 2.3. Physical Reasoning in Closed and Open Sets

Benchmarks such as PhysBench (Chow et al., 2025), PhysReason (Zhang et al., 2025b) and MMMU (Yue et al., 2024; 2025) have significantly advanced the evaluation of physical knowledge. However, these tasks are frequently presented in structured and closed-set formats where necessary variables are usually explicitly provided.

In contrast, authentic physical reasoning constitutes an open-set inference problem. This process requires autonomous discovery of unmentioned support evidence, such as noticing floor unevenness to determine stability (Li et al., 2026a). Consequently, existing benchmarks test the processing of physical rules (Xu et al., 2025) but fail to evaluate the acquisition of the visual evidence required to apply those rules.

## 3. The Visual Implicit Reasoning Deficit

We posit that the current limitations of VLMs stem primarily from a specific deficit in implicit visual reasoning rather than a general lack of cognitive capacity. In this section we formalize the distinction between explicit and implicit visual reasoning and map the current capabilities of VLMs onto a cognitive quadrant to isolate this specific deficit.

### 3.1. Formalizing Visual Implicit Reasoning

To understand the deficit, we mathematically characterize the reasoning process. Let a VLM $M$ take an image $I$ and a text query $Q$ to produce an answer $A$.

**Explicit Reasoning.** In the dominant paradigm of current training, $Q$ explicitly contains pointers to the necessary visual evidence $E$. For example, if $Q$ is *"Is the red cable connected to the battery?"*, the attention mechanism is linguistically guided to the specific regions of the red cable and battery. The task is reduced to a verification problem:

$$A \leftarrow M(I, Q_{\text{explicit}}) \qquad (1)$$

Here, the model's role is passive because it executes a predefined visual search plan provided by the user.

**Implicit Reasoning.** In contrast, implicit reasoning presents an under-specified query where the necessary $E$ is unmentioned. For example, $Q$ might simply be *"What is the diameter of this badge?"*.

To answer this question accurately, the model needs to autonomously identify and utilize implicit visual information in the image, such as a coin with a known diameter. This involves a coherent two-stage reasoning process. First, the model executes a *Plan* phase, leveraging prior world knowledge $K$ to formulate a specific search intent from $Q$. It then performs an autonomous *Search* operation, examining $I$ guided by this intent to uncover relevant but unmentioned evidence $E$.

$$E \leftarrow Search\big(I, Plan(Q, K)\big) \qquad (2)$$

Upon the successful retrieval of $E$, the cognitive task transitions from open-ended exploration to well-founded reasoning. The model is expected to integrate this discovered

visual context with the original query to derive the final conclusion. This ultimate inference step is formalized as:

$$A \leftarrow M(E, Q) \qquad (3)$$

**The Deficit.** We define the *Visual Implicit Reasoning Deficit* as a critical breakdown in the autonomous retrieval phase defined in Equation (2). When the necessary $E$ is not explicitly named in $Q$, the model fails to initiate the search mechanism $\text{Plan}(Q, K)$. Instead of treating the image as a field of potential evidence to be explored, the model tends to restrict its visual attention primarily to the entities explicitly mentioned in the text. Consequently, the reasoning process degenerates from the grounded deduction of Equation (3) to a constrained inspection of the target object:

$$A \leftarrow M(I|_{Q_{\text{target}}}, Q) \qquad (4)$$

where $I|_{Q_{\text{target}}}$ denotes the visual content restricted solely to the target object explicitly named in $Q$.

In this state, the model exhibits *attention tunneling*: it accurately processes the pixels of the object mentioned in the prompt (Visual Capacity) but treats the surrounding context, which contains the crucial unmentioned evidence, as irrelevant background. Lacking the retrieved context $E$, the model cannot construct a valid physical argument and often defaults to parametric hallucinations.

### 3.2. The Missing Quadrant: Autonomous Information Retrieval

To further contextualize this deficit, we propose a taxonomy of visual-language tasks structured along two axes. The first axis is *Information Availability* distinguishing between explicit and implicit inputs while the second is *Cognitive Demand* distinguishing between recognition and reasoning. As illustrated in Figure 1, this framework divides the landscape into four operational quadrants.

Quadrant **I** represents explicit recognition where targets are named and the goal is simple identification. In this domain current models exhibit high visual capacity. Quadrant **II** covers explicit reasoning typified by mathematical benchmarks where models perform well because the textual query provides a guided reasoning path. Quadrant **III** involves implicit perception which addresses passive quality control issues such as object hallucination but does not require active evidence seeking. Quadrant **IV** is defined as autonomous information Retrieval where visual evidence is decisive but entirely unmentioned in the prompt.

The critical gap lies within Quadrant IV which we identify as the *Missing Quadrant*. Unlike explicit tasks success here requires the model to autonomously deduce that spe-

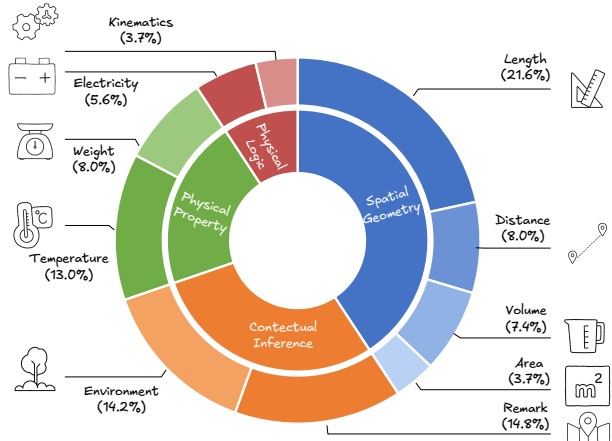

*Figure 3.* **Statistics of categories and tasks in V-IRD.**

cific unnamed features need to be retrieved to answer the high-level query. Current VLMs function as lazy readers operating effectively only when told where to look but failing fundamentally in this domain. They lack the visual agency to transform a high-level goal into a low-level visual search operation. This deficit explains the paradox where a model can perfectly describe a specific visual defect if explicitly asked yet confidently overlooks the same defect when making a holistic safety judgment simply because it never autonomously initiated the search for evidence.

## 4. Empirical Analysis: Boundaries and Mechanisms of Implicit Visual Reasoning in VLMs

To validate our hypothesis of the *Visual Implicit Reasoning Deficit*, we designed a comprehensive experimental framework encompassing mainstream VLMs shown in Figure 2. Rather than relying on large-scale automated benchmarks, which may contain statistical biases, we adopt a targeted evaluation strategy. Our experiments are structured logically to peel back the layers of the deficit: first establishing capacity boundaries, then exposing the lack of agency, and finally analyzing the cognitive breakdown mechanisms.

### 4.1. The V-IRD Benchmark

Current mainstream benchmarks like PhysBench (Chow et al., 2025) typically structure prompts with direct pointers, such as asking *"What is the color of the leftmost spectrum in the picture?"*. Recent works on visual reasoning like Point-It-Out (Xue et al., 2025) further highlight that models rely heavily on text-conditioned attention or visual markers to locate relevant regions. These prompts effectively leak the solution path. By explicitly naming the supporting evidence, these benchmarks evaluate the *verification capacity* of the

*Table 1.* **Validation of prerequisite capabilities.** The accuracy (%) of Visual Recognition and Parametric Knowledge is reported to ensure models possess the foundational knowledge required for subsequent reasoning tasks.

| Model Category | Task | |
|---|---|---|
| | Visual Recognition % | Parametric Knowledge % |
| Lightweight ($<$7B) | 99.74 | 74.81 |
| Medium-Scale (7-30B) | 100.00 | 87.01 |
| Large-Scale (30-80B) | 99.55 | 90.91 |
| Closed-source Models | 100.00 | 96.00 |

model rather than its *search agency*.

**Benchmark Construction and Taxonomy.** We adopted a multi-source collection strategy to guarantee visual diversity, sourcing images from manual photography, web crawling, and AI generation. As illustrated in Figure 3, we structured the V-IRD benchmark into a hierarchical taxonomy spanning four core domains to ensure comprehensive cognitive coverage. **Spatial Geometry** constitutes the largest portion (41%), focusing on precise metrology tasks such as *Length*, *Distance*, *Volume*, and *Area*. **Contextual Inference** (29%) challenges the model to deduce abstract information like *Environment* and *Remark*. The remaining data comprises **Physical Properties** (21%), covering *Temperature* and *Weight*, and **Physical Logic** (9%), which involves *Electricity* and *Kinematics*. In addition, we conduct a rigorous physical consistency check. We manually screened all images, especially those generated by AI, to remove logical errors or physically impossible content. This process guarantees that every visual clue is realistic and reliable.

**The Target-Exclusive Prompting Strategy.** To ensure that performance stems solely from intrinsic visual agency rather than instruction-following, we implement a strict Target-Exclusive Strategy. Unlike traditional visual prompting techniques that guide attention via spatial referents or bounding boxes, our prompts are rigorously sanitized to mention *only* the semantic target subject (*e.g.*, How tall is this bottle?). We explicitly forbid any textual references to background reference objects such as coins, environmental markers, or relative positions. This constraint forces the model to independently realize that unmentioned visual information is critical to the solution, effectively turning the evaluation into a pure test of *Autonomous Visual Discovery*.

### 4.2. Evaluation Metrics

To evaluate performance across different task types, we employ specific criteria to define prediction correctness.

**1) Discrete Tasks.** For reasoning tasks involving categorical outcomes such as logical reasoning and environment inference, standard Accuracy (ACC) is employed. A prediction is considered correct only if the predicted category exactly matches the ground-truth label. Mathematically, for a prediction $\hat{c}$ and ground truth $c$, the scoring function is:

$$\text{ACC} = \mathbb{I}(\hat{c} = c) \tag{5}$$

where $\mathbb{I}(\cdot)$ is the indicator function, which equals 1 if the condition is met and 0 otherwise.

**2) Continuous Tasks.** For estimation tasks as geometric scaling and physical properties, Threshold Accuracy ($\text{ACC}_\delta$) is employed. Unlike regression metrics that average continuous errors, we adopt a strict binary success criterion. A prediction is considered correct only if its relative error falls within a fixed threshold $\delta$:

$$\text{ACC}_\delta = \mathbb{I}\left( \frac{|y - \hat{y}|}{y} \leq \delta \right) \tag{6}$$

In this case, $y$ represents the ground-truth value, and $\hat{y}$ is the predicted value. Note that for edge cases where the ground truth is zero (*e.g.*, $0°$C), the metric degenerates to standard ACC, requiring an exact match. To evaluate reasoning precision at different granularities, we adopt three distinct thresholds $\delta \in \{0.05, 0.10, 0.20, 0.30\}$ (representing 5%, 10%, 20% and 30% tolerance margins). If the error exceeds the given threshold, the score is 0. This strict evaluation regime penalizes vague guesses and distinguishes high-precision visual measurements from coarse approximations.

### 4.3. Verification of Explicit Capability

Before studying implicit reasoning, the factor of **lack of knowledge** needs to be excluded. We need to verify whether the failures stem from a lack of agency or simply because the model lacks the required fundamental knowledge.

We extracted the core **Implicit Information Components** underpinning the V-IRD benchmark, which broadly encompass key visual reference objects (e.g., standard coins, ID cards), fundamental physical rules, and environmental information. To ensure a comprehensive evaluation, we categorized the models into four distinct groups based on parameter size and accessibility: **Lightweight** ($<$ 7B), **Medium-Scale** ($7 - 30$B), **Large-Scale** ($30 - 80$B), and **Closed-source Models**. We structured the Explicit Capability Probe into two distinct verification stages. The first stage evaluates **Visual Recognition** to determine if the model can accurately identify the implicit objects from the visual input. The second stage evaluates **Parametric Knowledge** by querying the model via text to confirm if it knows the specific physical attributes corresponding to these objects. This process effectively acts as a unit test for the model's knowledge base, isolating its foundational capabilities from its reasoning agency.

**Results.** As presented in Table 1, the evaluation of prerequisite capabilities reveals that visual recognition has reached saturation, with accuracy exceeding 99.55% across all categories and achieving a perfect 100.00% in Medium-Scale

*Table 2.* **Main results on V-IRD.** We benchmark a broad range of VLMs across four domains and ten fine-grained tasks. Each cell reports accuracy (%) under relative-error thresholds of 5% and 10%, formatted as (ACC$_{5\%}$, ACC$_{10\%}$). The **Average** column gives the mean accuracy across all ten tasks at the two thresholds. Within each model group, the best score per column is highlighted in **bold**. Across all VLM groups, the top-3 entries per column are shaded with decreasing intensity: 1st , 2nd , 3rd .

| Model | Spatial Geometry | | | | Physical Properties | | Physical Logic | | Contextual Inference | | Average |
|---|---|---|---|---|---|---|---|---|---|---|---|
| | Length | Area | Volume | Distance | Temp. | Weight | Electricity | Kinematics | Remark | Env. | |
| *Open-source VLMs : < 7B* | | | | | | | | | | | |
| InternVL3.5-1B (Wang et al., 2025a) | ( 7.14, 10.00) | ( 0.00, 16.67) | ( 0.00, 4.17) | **(11.54, 34.62)** | (54.76, 59.52) | (34.62, 34.62) | (11.11, 11.11) | **(50.00, 50.00)** | (16.67, 16.67) | (43.48, 43.48) | (22.93, 28.08) |
| InternVL3.5-2B (Wang et al., 2025a) | ( 0.00, 8.57) | ( 0.00, 8.33) | ( 8.33, 12.50) | ( 0.00, 7.69) | (69.05, 76.19) | (34.62, 34.62) | (38.89, 38.89) | (33.33, 33.33) | (25.00, 33.33) | (56.52, 56.52) | (26.57, 31.00) |
| InternVL3-1B (Zhu et al., 2025) | ( 4.29, 8.57) | ( 8.33, 8.33) | ( 4.17, 12.50) | ( 3.85, 19.23) | (28.57, 30.95) | (26.92, 26.92) | (11.11, 11.11) | (41.67, 41.67) | (27.08, 31.25) | (52.17, 52.17) | (20.82, 24.27) |
| InternVL3-2B (Zhu et al., 2025) | ( 7.14, 20.00) | ( 0.00, 0.00) | **(12.50, 20.83)** | ( 3.85, 19.23) | **(78.57, 85.71)** | (42.31, 42.31) | (55.56, 55.56) | (41.67, 58.33) | (52.08, 52.08) | (54.35, 54.35) | **(34.80, 40.84)** |
| Qwen3-VL-2B (Yang et al., 2025) | ( 8.57, 15.71) | ( 8.33, 16.67) | ( 4.17, 8.33) | ( 3.85, 11.54) | (71.43, 71.43) | (53.85, 57.69) | (33.33, 33.33) | (33.33, 33.33) | (39.58, 47.92) | (54.35, 54.35) | (31.08, 35.03) |
| Qwen3-VL-4B (Yang et al., 2025) | **(20.00, 31.43)** | ( 0.00, 8.33) | ( 0.00, 16.67) | ( 0.00, 0.00) | (71.43, 71.43) | **(76.92, 76.92)** | (38.89, 50.00) | (33.33, 41.67) | (25.00, 25.00) | **(67.39, 67.39)** | (33.30, 38.89) |
| Qwen2.5-VL-3B (Bai et al., 2025) | ( 8.57, 20.00) | **(16.67, 16.67)** | ( 4.17, 16.67) | (11.54, 26.92) | (71.43, 71.43) | (38.46, 42.31) | (27.78, 27.78) | (16.67, 25.00) | (35.42, 35.42) | (60.87, 60.87) | (29.16, 34.31) |
| *Open-source VLMs : 7 – 8B* | | | | | | | | | | | |
| Qwen3-VL-8B (Yang et al., 2025) | (10.00, 22.86) | ( 0.00, 8.33) | **(12.50, 20.83)** | (11.54, 19.23) | (90.48, 92.86) | (61.54, 65.38) | **(44.44, 44.44)** | (41.67, 41.67) | (35.42, 37.50) | **(86.96, 86.96)** | **(39.45, 44.01)** |
| Qwen2.5-VL-7B (Bai et al., 2025) | ( 7.14, 20.00) | **( 8.33, 16.67)** | ( 0.00, 8.33) | ( 3.85, 23.08) | (52.38, 52.38) | **(65.38, 69.23)** | (27.78, 27.78) | ( 8.33, 8.33) | (29.17, 31.25) | (65.22, 65.22) | (26.76, 32.23) |
| InternVL3.5-8B (Wang et al., 2025a) | ( 7.14, 15.71) | ( 0.00, 0.00) | ( 8.33, 16.67) | ( 7.69, 11.54) | (90.48, 90.48) | (50.00, 50.00) | (33.33, 33.33) | (50.00, 50.00) | (43.75, 45.83) | (71.74, 71.74) | (36.25, 38.53) |
| InternVL3-8B (Zhu et al., 2025) | ( 4.29, 11.43) | ( 0.00, 33.33) | ( 0.00, 25.00) | ( 3.85, 3.85) | (50.00, 59.52) | (42.31, 42.31) | (22.22, 22.22) | **(58.33, 58.33)** | **(60.42, 66.67)** | (73.91, 73.91) | (31.53, 39.66) |
| LLaVA-OV-7B (Li et al., 2024a) | **(10.00, 27.14)** | ( 0.00, 0.00) | ( 4.17, 12.50) | **(11.54, 23.08)** | (59.52, 59.52) | (46.15, 46.15) | ( 5.56, 5.56) | (33.33, 50.00) | (43.75, 47.92) | (71.74, 71.74) | (28.58, 34.36) |
| *Open-source VLMs : 10 – 30B* | | | | | | | | | | | |
| InternVL3.5-14B (Wang et al., 2025a) | ( 7.14, 18.57) | ( 0.00, 8.33) | ( 8.33, 12.50) | ( 3.85, 15.38) | (80.95, 80.95) | (50.00, 50.00) | (27.78, 27.78) | (83.33, 83.33) | (66.67, 70.83) | (69.57, 69.57) | (39.76, 43.73) |
| InternVL3-14B (Zhu et al., 2025) | **(12.86, 22.86)** | **( 8.33, 8.33)** | **(12.50, 41.67)** | (23.08, 34.62) | **(85.71, 85.71)** | **(53.85, 53.85)** | **(33.33, 33.33)** | (50.00, 50.00) | (50.00, 54.17) | **(73.91, 73.91)** | **(40.36, 45.84)** |
| *Open-source VLMs : 30 – 70B* | | | | | | | | | | | |
| InternVL3-38B (Zhu et al., 2025) | (11.43, 20.00) | ( 0.00, 0.00) | **(12.50, 16.67)** | (11.54, 23.08) | (90.48, 95.24) | (69.23, 69.23) | (27.78, 33.33) | (58.33, 58.33) | (62.50, 66.67) | (78.26, 78.26) | (42.20, 46.08) |
| InternVL3.5-38B (Wang et al., 2025a) | (12.86, 24.29) | ( 0.00, 0.00) | ( 4.17, 16.67) | (15.38, 34.62) | (90.48, 90.48) | (69.23, 69.23) | (50.00, 55.56) | (66.67, 66.67) | (58.33, 62.50) | (76.09, 76.09) | **(44.32, 49.61)** |
| Qwen2.5-VL-32B (Bai et al., 2025) | (11.43, 24.29) | **( 8.33, 16.67)** | ( 0.00, 8.33) | ( 0.00, 3.85) | (76.19, 76.19) | **(73.08, 80.77)** | (44.44, 44.44) | (50.00, 50.00) | (47.92, 52.08) | (69.57, 69.57) | (38.10, 42.62) |
| Qwen3-VL-32B (Yang et al., 2025) | **(15.71, 22.86)** | ( 0.00, 16.67) | **(12.50, 16.67)** | ( 7.69, 11.54) | **(92.86, 95.24)** | (73.08, 73.08) | (44.44, 44.44) | (50.00, 58.33) | (54.17, 54.17) | **(84.78, 84.78)** | (43.52, 47.78) |
| *Open-source VLMs : 70 – 80B* | | | | | | | | | | | |
| LLaVA-NEXT-72B (Li et al., 2024b) | (15.71, 35.71) | ( 0.00, 0.00) | ( 0.00, 12.50) | ( 0.00, 3.85) | (73.81, 73.81) | (38.46, 38.46) | (11.11, 11.11) | (33.33, 33.33) | (14.58, 16.67) | (54.35, 54.35) | (24.14, 27.98) |
| LLaVA-OV-72B (Li et al., 2024a) | **(17.14, 30.00)** | ( 0.00, 0.00) | ( 8.33, 16.67) | ( 0.00, 11.54) | (88.10, 90.48) | (38.46, 38.46) | (61.54, 61.54) | (66.67, 66.67) | (47.92, 47.92) | (78.26, 78.26) | (40.68, 44.20) |
| Qwen2.5-VL-72B (Bai et al., 2025) | (11.43, 21.43) | **(16.67, 25.00)** | ( 0.00, 0.00) | ( 3.85, 3.85) | (80.95, 80.95) | **(84.62, 88.46)** | (50.00, 50.00) | (50.00, 50.00) | (60.42, 66.67) | (78.26, 78.26) | (43.62, 46.46) |
| InternVL3-78B (Zhu et al., 2025) | (14.29, 25.71) | ( 8.33, 25.00) | ( 8.33, 20.83) | (11.54, 23.08) | (90.48, 90.48) | (69.23, 69.23) | (22.22, 22.22) | (66.67, 66.67) | (66.67, 70.83) | (84.78, 84.78) | (44.25, 49.88) |
| *Open-source VLMs : > 200B* | | | | | | | | | | | |
| Qwen3-VL-235B (Yang et al., 2025) | (14.29, 27.14) | (16.67, 25.00) | ( 0.00, 16.67) | ( 7.69, 7.69) | (90.48, 97.62) | (76.92, 76.92) | (44.44, 50.00) | (66.67, 66.67) | (62.50, 62.50) | (84.78, 84.78) | (46.44, 51.50) |
| *Proprietary VLMs* | | | | | | | | | | | |
| GPT-5.2 (Singh et al., 2025) | (14.29, 27.14) | (16.67, 16.67) | ( 8.33, 16.67) | ( 0.00, 3.85) | (92.86, 100.00) | (80.77, 84.62) | (50.00, 55.56) | (58.33, 58.33) | (75.00, 77.08) | (78.26, 78.26) | (47.45, 51.82) |
| Claude-Sonnet-4.5 (Anthropic, 2025) | (27.14, 44.29) | (16.67, 33.33) | (29.17, 37.50) | ( 0.00, 3.85) | (88.10, 88.10) | (73.08, 73.08) | (38.89, 44.44) | (66.67, 75.00) | (72.92, 75.00) | (86.96, 86.96) | (49.96, 56.15) |
| Claude-Sonnet-4.5-Thinking (Anthropic, 2025) | (25.71, 41.43) | (16.67, 16.67) | (12.50, 16.67) | ( 3.85, 15.38) | (85.71, 88.10) | (69.23, 69.23) | (33.33, 33.33) | (83.33, 83.33) | (75.00, 79.17) | (91.30, 91.30) | (49.66, 53.46) |
| Gemini-3-Flash (Google DeepMind, 2025) | (35.71, 58.57) | (16.67, 41.67) | (29.17, 33.33) | ( 3.85, 19.23) | (95.24, 100.00) | (65.38, 69.23) | (66.67, 66.67) | (75.00, 83.33) | (79.17, 79.17) | (84.78, 84.78) | (55.16, 63.60) |
| Gemini-3-Pro (Google DeepMind, 2025) | (44.29, 54.29) | (25.00, 50.00) | (16.67, 29.17) | (11.54, 11.54) | (95.24, 97.62) | (80.77, 80.77) | (66.67, 66.67) | (75.00, 83.33) | (77.08, 77.08) | (91.30, 91.30) | (58.36, 64.18) |
| *Human reference* | | | | | | | | | | | |
| **Human** | (40.00, 60.00) | (16.67, 25.00) | (50.00, 50.00) | (38.46, 46.15) | (100.00, 100.00) | (92.31, 92.31) | (66.67, 66.67) | (83.33, 83.33) | (83.33, 83.33) | (91.30, 91.30) | (66.21, 69.81) |

and Closed-source models, indicating robust perceptual systems. Concurrently, Parametric Knowledge demonstrates a positive correlation with model scale, where performance steadily improves from a baseline of 74.81% in Lightweight models to 87.01% and 90.91% in Medium-Scale and Large-Scale models respectively, culminating in a peak of 96.00% for Closed-source architectures. These findings confirm that the essential atomic knowledge required for V-IRD tasks is already encoded within these systems, thereby isolating the variable of interest and verifying that subsequent failures in implicit reasoning stem from an agency deficit rather than a fundamental lack of capability.

### 4.4. Core Evaluation: The Deficit of Active Visual Reasoning

**Experimental Settings.** We evaluated standard VLMs on the V-IRD benchmark using the *Target-Exclusive Strategy* in Table 2. In this setting, prompts explicitly request the final target value (Target $T$) but intentionally omit any mention of the available visual evidence (Implicit Information $I$). This forces the model to autonomously discover and utilize

the visual context. Performance is measured using ACC$_\delta$ at strict ($\delta = 5\%$) and relaxed ($\delta = 10\%$) margins for continuous tasks, and standard Accuracy for discrete tasks.

**Severe Collapse in Spatial Geometry.** The results reveal a significant performance divergence, with the most catastrophic failure observed in *Spatial Geometry*. While models demonstrated high precision in explicit pre-experiments (where reference objects were named), their performance declined significantly under the target-exclusive setting. For instance, even at the relaxed threshold ($\delta = 10\%$), most models struggled to achieve meaningful accuracy. This degradation suggests that without explicit text pointing to reference objects, models frequently struggle to actively look for them, which leads to hallucinations based on prior training distributions rather than situated visual measurement.

**Performance in Physical Tasks and Contextual Inference.** Other domains exhibited varying degrees of fragility. *Physical Properties* and *Physical Logic* maintained relatively good performance where visual components were salient. Conversely, in *Contextual Inference* tasks requiring context

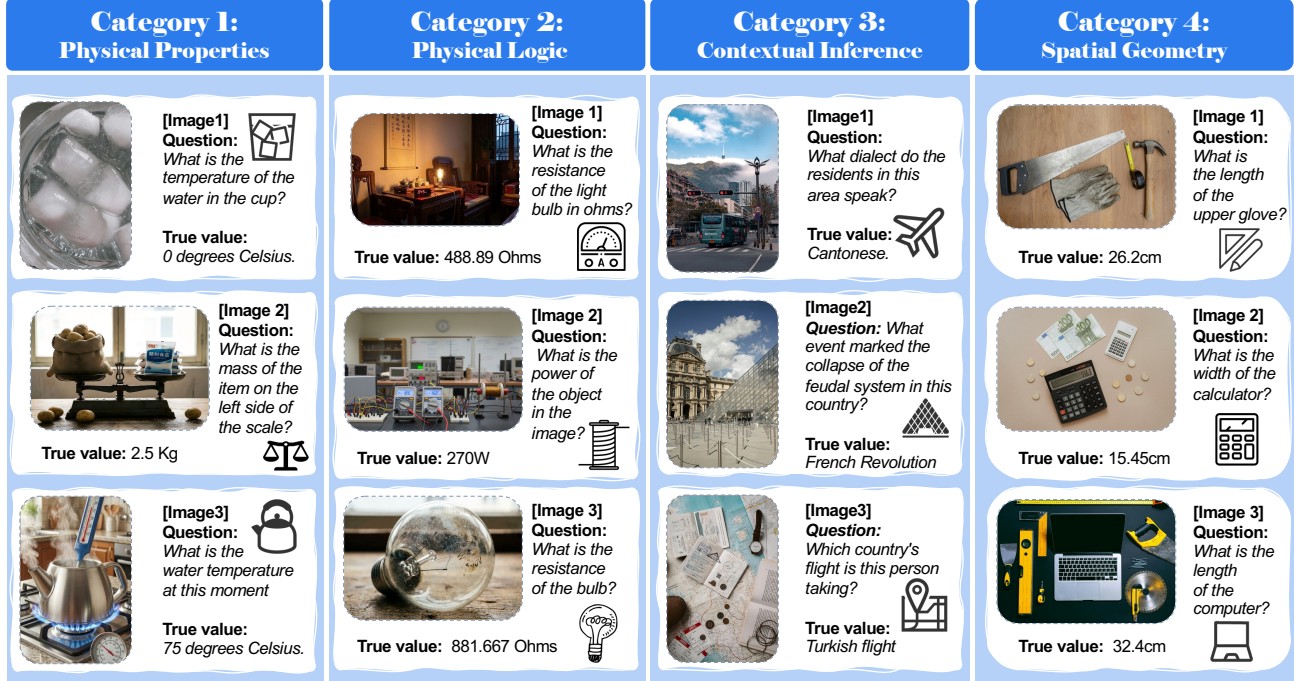

*Figure 4.* **Representative instances of implicit reasoning.** Category 1: Physical Properties infers intrinsic attributes, ranging from thermodynamic states to mass equivalence calculations based on conservation principles. Category 2: Physical Logic applies specific physical laws to analyze functional systems. Category 3: Contextual Inference deduces non-visual contexts from environmental clues. Category 4: Spatial Geometry performs precise metrology using reference objects. Success in these domains requires visual agency, which is the ability to actively retrieve unmentioned evidence for high-level queries.

deduction, models frequently ignored background evidence in favor of generic foreground features. Overall, Closed-source models generally outperformed Open-source models across these domains, exhibiting stronger robustness in active retrieval, although the visual agency deficit remains a widespread challenge across current architectures.

### 4.5. Probing Agency Deficit in Visual Reasoning

To further probe the robustness of the reasoning agency, we conducted a focused qualitative analysis on a curated set of 10 complex samples. These images are characterized by high information density with multiple potential visual references, yet contain sparse valid cues specifically applicable to the reasoning target. We selected the top-performing models at the 5% threshold across different scales (InternVL3-2B, Qwen3-VL-8B, InternVL3-14B, InternVL3.5-38B, InternVL3-78B, and Gemini-3-pro) and instructed them to generate explicit Chain-of-Thought (Wei et al., 2022) sequences to solve these tasks. A prediction is considered correct if the relative error falls within a 10% threshold ($\delta < 10\%$). By analyzing their generated traces, we pinpoint exactly where the cognitive chain breaks. We formulated a hierarchical taxonomy to categorize failures into three sequential stages:

**Stage I: Active Discovery Failure.** The model provides a detailed description of the clutter but fails to acknowledge the presence of the specific implicit information required for the task.

**Stage II: Valuation and Selection Failure.** The model explicitly notices the valid visual evidence but fails to establish a logical connection with the target. In these information-rich scenarios, the model often treats the critical cue as irrelevant background noise, overwhelmed by other salient but non-functional objects.

**Stage III: Logical Calculation Failure.** The model successfully bridges the anchor and the target but fails at the stage of physical modeling or numerical computation.

**Analysis Results.** As shown in Table 3, the quantitative results reveal a decisive skew towards early-stage perceptual deficits. On average, 75.82% of failures are classified as Stage I, indicating that models predominantly fail to perceive the implicit cues entirely. This is particularly pronounced in smaller models, where Stage I errors reach 90%. Stage II accounts for an additional 14.42%, where evidence is noticed but treated as noise. Consequently, the combined *Agency Deficit* (Stage I & II) constitutes over 90% of all failures. In contrast, only 9.76% of errors result from Stage III calculation failures. Notably, smaller models exhibit 0% logic failure simply because they rarely survive the discov-

*Table 3.* **Quantitative analysis of failure stages.** Errors are decomposed into *Agency Deficit* (Stage I: Discovery, Stage II: Association) and *Capacity Deficit* (Stage III: Logic). Results confirm that reasoning is primarily bottlenecked by an inability to actively find the correct implicit information, which accounts for the vast majority of failures and significantly outweighs logic-based errors.

| Model | Accuracy (%) | Agency Deficit (%) | | Capacity Deficit (%) |
|---|---|---|---|---|
| | | I: Discovery | II: Association | III: Logic |
| InternVL3-2B | 0.00 | 90.00 | 10.00 | 0.00 |
| Qwen3-VL-8B | 10.00 | 88.89 | 11.11 | 0.00 |
| InternVL3-14B | 10.00 | 88.89 | 11.11 | 0.00 |
| InternVL3.5-38B | 0.00 | 70.00 | 20.00 | 10.00 |
| InternVL3-78B | 30.00 | 57.14 | 14.29 | 28.57 |
| Gemini-3-Pro | 50.00 | 60.00 | 20.00 | 20.00 |
| **Average** | **16.67** | **75.82** | **14.42** | **9.76** |

ery phase to attempt calculation. Even for the strongest model (Gemini-3-pro), the *Agency Deficit* remains at 80%, confirming that the primary bottleneck is not calculation capacity, but the agency to initiate search.

### 4.6. Diagnostic Experiment: Explicit Information Injection

To conclusively verify that the performance collapse stems from an agency deficit rather than capability limitations, we conducted a gradient-based information injection experiment. We used the same model and data configurations as in the previous experiment, employing the same carefully curated high-information-density samples and representative model set.

We designed a four-stage protocol to observe performance evolution as visual planning is offloaded to the prompt. The experiment starts at Level 0 (Implicit Baseline) with original underspecified instructions. We then introduce Level 1 (object awareness) to explicitly prompt attention to specific references, followed by Level 2 (attribute awareness) which specifies the physical attributes to inspect. Finally, Level 3 (oracle guidance) provides ground truth data for comparison and reduces the task to pure logical calculation. Consistent with the previous experiment, a prediction is considered correct if the relative error falls within a 10% threshold ($\delta < 10\%$).

**Results and Implications.** As shown in Figure 5, for large-scale models, the highly explicit physical information injected into the prompts may conflict with their extensive pre-trained knowledge. Such models generally depend on robust parametric priors. Therefore, discrepancies between the provided physical cues and their inherent knowledge may induce reasoning conflicts, leading to stagnant or diminished performance. Conversely, medium-scale models demonstrate an improvement. This positive trend indicates that explicit external guidance mitigates their intrinsic deficit in active visual exploration to a certain degree. Meanwhile,

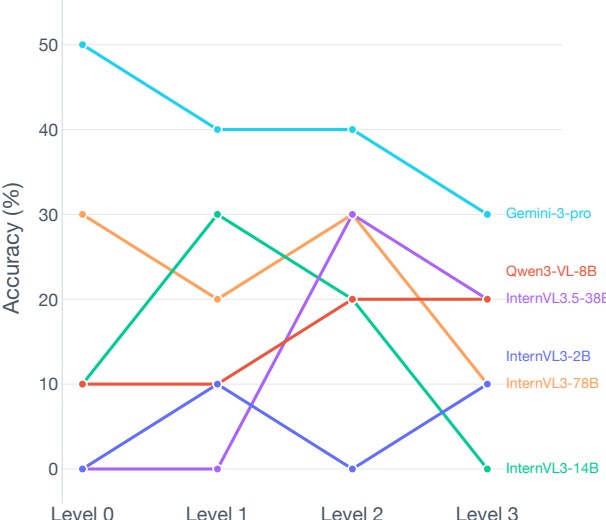

*Figure 5.* **Performance trends across difficulty levels (level 0-3).** The scale-dependent effects of explicit information injection on model performance. While medium-scale models benefit from explicit guidance, large-scale and small-scale models exhibit stagnant or diminished performance due to conflicts with parametric priors and limited baseline capacity, respectively.

the performance of small-scale models remains relatively static. This limitation is presumably attributable to their restricted baseline capacity, which impedes their ability to process and integrate complex visual cues into the final reasoning procedures.

## 5. Alternative Views

A prevalent perspective in recent VLMs research is that the observed limitations in visual implicit reasoning are primarily a consequence of insufficient scale or suboptimal elicitation, rather than a fundamental shortcoming of current models. From this viewpoint, increasing model capacity, training data, or prompt specificity should naturally resolve the reported failures.

**Scaling Capacity vs. Scaling Agency.** One prominent hypothesis argues that implicit reasoning is an emergent capability that will reliably manifest as models continue to scale (Wei et al., 2022). Indeed, our results confirm that scaling consistently improves visual and physical knowledge, suggesting substantial gains in representational capacity. However, we observe that improvements in implicit visual reasoning are markedly slower and less stable (Huang et al., 2025). This divergence indicates that scaling preferentially enhances *what* a model knows, but does not guarantee *when* or *why* that knowledge is autonomously deployed. In this sense, scaling capacity does not equate to scaling agency.

**Prompt Sensitivity and Evaluation Intent.** Another alternative explanation attributes the observed failures to prompt design or evaluation artifacts, arguing that more explicit instructions or chain-of-thought prompts (Wei et al., 2022; Zhang et al., 2024) would elicit stronger reasoning behavior (Li et al., 2026b). While this interpretation is plausible, it overlooks the specific intent of our evaluation. Our focus on implicit reasoning is motivated by whether models can proactively uncover and exploit relevant information in an image without rich, task-specific guidance. Prompt minimalism is therefore a deliberate probe of default model behavior, rather than a limitation of the evaluation setup. If reasoning only emerges when explicitly requested, it reflects a gap between possessing knowledge and deploying it by default.

Scaling and prompting improve guided performance but do not enable autonomous visual reasoning. Addressing the failure to initiate reasoning in implicit settings necessitates solutions rooted in fundamental model architecture, rather than simple scale or guidance.

## 6. Discussion and Conclusion

This paper establishes a formal framework for visual implicit reasoning and reveals a fundamental limitation of current VLMs: a gap between passive recognition and the active agency required for genuine visual understanding. Both theoretical analysis and evidence from V-IRD show that contemporary models function primarily as probabilistic semantic retrievers rather than grounded visual reasoners.

Our analysis further indicates that this limitation is not due to missing atomic knowledge, but to a pronounced agency deficit. When faced with implicit instructions, models abandon situated visual measurement and default to internal memory. These findings suggest that scaling model size alone is insufficient; progress instead requires training objectives that promote autonomous visual discovery and active perception.

## Acknowledgement

This work was funded by the National Natural Science Foundation of China (Grant No. 62571379) and the Hubei Provincial Key Research and Development Program (Grant No. 2024BAB050). The numerical calculations in this paper have been done on the supercomputing system in the Supercomputing Center of Wuhan University.

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

# A. Supplementary Quantitative Analysis

To systematically investigate whether the observed performance deficits arise from strict metric precision requirements or a fundamental lack of visual grounding, we analyze model performance under relaxed error tolerances ($\delta \in \{20\%, 30\%\}$), as detailed in Table 4. This ablation effectively isolates *calibration sensitivity* from *reasoning capability*. The results demonstrate a distinct performance bifurcation that implies different error mechanisms across domains.

In semantic-centric tasks, specifically *Physical Properties* and *Contextual Inference*, relaxing the error threshold yields immediate and substantial performance gains. Proprietary models such as GPT-5.2 and Gemini-3-Pro approach ceiling performance (near 100% accuracy) even at the 20% threshold. This rapid saturation suggests that failures in these domains are largely attributable to minor calibration variances rather than flawed reasoning logic.

Conversely, *Spatial Geometry* tasks exhibit a structural failure mode that is resistant to threshold relaxation. Even with a generous 30% tolerance, performance in *Volume* and *Distance* estimation remains critically low. For instance, InternVL3-78B attains only $\sim 33\%$ accuracy on Volume estimation at the 30% threshold, in sharp contrast to its 95% accuracy on Temperature estimation. This persistent stagnation suggests that the error in spatial tasks is, to a certain degree, rooted in foundational deficiencies. Instead of actively establishing a visual reference scale, models revert to unimodal parametric priors, resulting in hallucinations that deviate fundamentally from the visual reality, a limitation that cannot be masked by simply loosening evaluation constraints.

# B. Inference Prompts

## B.1. Universal System Prompt

All models are conditioned with a unified system instruction.

---

**System Instruction Prompt**

**Role:** You are a multimodal AI assistant specializing in precise physical reasoning and geometric estimation.

**Core Directive:** Answer the user's question by deriving the result strictly from the visual content provided. You must first provide an explicit reasoning process explaining how you calculated or deduced the answer, grounded solely in observable pixel data.

**Negative Constraints:**
1. Do NOT rely on generic parametric knowledge (e.g., "standard sizes") if it conflicts with the visual evidence.
2. Do NOT output ranges (e.g., "20-30cm") or uncertainty terms (e.g., "approx", "maybe").
3. Do NOT provide verbose conclusions; output strictly the requested value.

**Output Schema:** You must output a single JSON object. The content of the "answer" field must start with the exact phrase "The answer is".

```
{ "reasoning":"Step-by-step derivation...", "answer":  "The answer is [Exact_Value]" }
```

---

## B.2. Domain-Specific Evaluation Prompts

We categorize the benchmark into four reasoning quadrants. For each instance, we employ a dual-query strategy to ablate the efficacy of explicit reasoning triggers.

---

**Q1. Physical Properties**
Evaluates the capacity to deduce invisible object states (e.g., thermal properties) via implicit secondary visual artifacts like condensation, rather than semantic recognition.

**Standard Prompt:** `"What is the temperature (°C) of the water in the picture?"`

**CoT Prompt:** `"What is the temperature (°C) of the water in the picture?  Please think step by step."`

---

*Table 4.* **Supplementary results on the V-IRD (20% and 30% thresholds).** We evaluate a wide range of VLMs across four distinct domains under relaxed constraints. The values in parentheses denote the Accuracy achieved under relative error thresholds of 20% and 30%, respectively. The **Average** column represents the mean accuracy across all tasks for these two thresholds. Within each model group, the best score per column is highlighted in **bold**. Across all VLM groups, the top-3 entries per column are shaded with decreasing intensity: 1st , 2nd , 3rd .

| Model | Spatial Geometry | | | | Physical Properties | | Physical Logic | | Contextual Inference | | Average |
|---|---|---|---|---|---|---|---|---|---|---|---|
| | Length | Area | Volume | Distance | Temp. | Weight | Electricity | Kinematics | Remark | Env. | |
| *Open-source VLMs : < 7B* | | | | | | | | | | | |
| InternVL3.5-1B (Wang et al., 2025a) | (11.43, 18.57) | (16.67, 16.67) | ( 4.17, 12.50) | **(34.62, 46.15)** | (59.52, 64.29) | (42.31, 50.00) | (11.11, 11.11) | (50.00, 66.67) | (16.67, 16.67) | (43.48, 43.48) | (29.00, 34.61) |
| InternVL3.5-2B (Wang et al., 2025a) | (20.00, 24.29) | (25.00, 25.00) | (12.50, 20.83) | ( 7.69, 26.92) | (76.19, 78.57) | (38.46, 50.00) | (38.89, 38.89) | (33.33, 58.33) | (33.33, 33.33) | (56.52, 56.52) | (34.19, 41.27) |
| InternVL3-1B (Zhu et al., 2025) | (14.29, 22.86) | (33.33, 33.33) | (12.50, 16.67) | (19.23, 23.08) | (35.71, 35.71) | (26.92, 38.46) | (11.11, 11.11) | (41.67, 41.67) | (33.33, 33.33) | (52.17, 52.17) | (28.03, 30.84) |
| InternVL3-2B (Zhu et al., 2025) | (25.71, 35.71) | ( 0.00, 0.00) | (20.83, 20.83) | (23.08, 30.77) | **(85.71, 85.71)** | (50.00, 57.69) | **(55.56, 61.11)** | **(58.33, 58.33)** | **(56.25, 60.42)** | (54.35, 54.35) | (42.98, 46.50) |
| Qwen3-VL-2B (Yang et al., 2025) | (27.14, 34.29) | (25.00, 25.00) | (20.83, 25.00) | (11.54, 23.08) | (73.81, 76.19) | (73.08, 76.92) | (38.89, 38.89) | (33.33, 50.00) | (47.92, 50.00) | (54.35, 54.35) | (40.59, 45.37) |
| Qwen3-VL-4B (Yang et al., 2025) | **(47.14, 55.71)** | (16.67, 16.67) | **(20.83, 29.17)** | ( 0.00, 0.00) | (73.81, 78.57) | **(92.31, 92.31)** | (50.00, 50.00) | (50.00, 50.00) | (25.00, 25.00) | **(67.39, 67.39)** | **(44.32, 46.48)** |
| Qwen2.5-VL-3B (Bai et al., 2025) | (42.86, 45.71) | **(41.67, 41.67)** | (16.67, 25.00) | (26.92, 30.77) | (76.19, 80.95) | (46.15, 53.85) | (33.33, 33.33) | (25.00, 25.00) | (35.42, 37.50) | (60.87, 60.87) | (40.51, 43.47) |
| *Open-source VLMs : 7 − 8B* | | | | | | | | | | | |
| Qwen3-VL-8B (Yang et al., 2025) | **(44.29, 55.71)** | (25.00, 33.33) | (20.83, 33.33) | (19.23, 19.23) | **(92.86, 95.24)** | (73.08, 80.77) | **(44.44, 44.44)** | (58.33, 58.33) | (37.50, 41.67) | **(86.96, 86.96)** | **(50.25, 54.90)** |
| Qwen2.5-VL-7B (Bai et al., 2025) | (38.57, 50.00) | (25.00, 41.67) | ( 8.33, 12.50) | **(23.08, 42.31)** | (52.38, 52.38) | **(76.92, 84.62)** | (27.78, 27.78) | (16.67, 16.67) | (33.33, 33.33) | (65.22, 65.22) | (36.73, 42.65) |
| InternVL3.5-8B (Wang et al., 2025a) | (41.43, 44.29) | (16.67, 33.33) | **(25.00, 29.17)** | (11.54, 15.38) | (90.48, 95.24) | (65.38, 69.23) | (38.89, 38.89) | (50.00, 75.00) | (52.08, 52.08) | (71.74, 71.74) | (54.22, 52.44) |
| InternVL3-8B (Zhu et al., 2025) | (28.57, 45.71) | **(33.33, 33.33)** | (25.00, 25.00) | (11.54, 23.08) | (61.90, 66.67) | (57.69, 57.69) | (22.22, 22.22) | **(83.33, 91.67)** | **(68.75, 72.92)** | (73.91, 76.09) | (46.63, 51.44) |
| LLaVA-OV-7B (Li et al., 2024a) | (42.86, 54.29) | ( 8.33, 16.67) | (12.50, 16.67) | (23.08, 26.92) | (59.52, 69.05) | (46.15, 57.69) | ( 5.56, 5.56) | (58.33, 66.67) | (52.08, 52.08) | (71.74, 71.74) | (38.02, 43.73) |
| *Open-source VLMs : 10 − 30B* | | | | | | | | | | | |
| InternVL3.5-14B (Wang et al., 2025a) | (41.43, 50.00) | (16.67, 33.33) | (12.50, 20.83) | (26.92, 34.62) | (80.95, 85.71) | (65.38, 73.08) | (27.78, 27.78) | (83.33, 100.00) | (70.83, 70.83) | (69.57, 69.57) | (49.54, 56.58) |
| InternVL3-14B (Zhu et al., 2025) | **(42.86, 54.29)** | ( 8.33, 8.33) | **(45.83, 45.83)** | **(46.15, 46.15)** | **(85.71, 85.71)** | **(73.08, 80.77)** | **(38.89, 38.89)** | (58.33, 91.67) | (62.50, 62.50) | **(73.91, 73.91)** | **(53.56, 58.81)** |
| *Open-source VLMs : 30 − 70B* | | | | | | | | | | | |
| InternVL3-38B (Zhu et al., 2025) | **(44.29, 61.43)** | (25.00, 33.33) | **(20.83, 37.50)** | **(38.46, 53.85)** | **(95.24, 95.24)** | (84.62, 92.31) | (33.33, 33.33) | (66.67, 83.33) | (66.67, 70.83) | (78.26, 78.26) | (55.34, 63.94) |
| InternVL3.5-38B (Wang et al., 2025a) | (45.71, 50.00) | (16.67, 41.67) | (20.83, 29.17) | (34.62, 34.62) | (90.48, 95.24) | **(92.31, 92.31)** | **(55.56, 55.56)** | (66.67, 100.00) | (64.58, 66.67) | (76.09, 80.43) | **(56.35, 64.57)** |
| Qwen2.5-VL-32B (Bai et al., 2025) | (40.00, 45.71) | (25.00, 41.67) | ( 8.33, 16.67) | (15.38, 15.38) | (78.57, 80.95) | (80.77, 92.31) | (44.44, 44.44) | (58.33, 75.00) | (54.17, 58.33) | (69.57, 69.57) | (47.46, 54.00) |
| Qwen3-VL-32B (Yang et al., 2025) | (34.29, 47.14) | **(33.33, 41.67)** | (16.67, 25.00) | (19.23, 26.92) | **(95.24, 95.24)** | (80.77, 96.15) | (44.44, 44.44) | (66.67, 75.00) | (56.25, 58.33) | **(84.78, 84.78)** | (53.17, 59.47) |
| *Open-source VLMs : 70 − 80B* | | | | | | | | | | | |
| LLaVA-NEXT-72B (Li et al., 2024b) | (52.86, 54.29) | (25.00, 33.33) | (12.50, 12.50) | (11.54, 15.38) | (73.81, 83.33) | (38.46, 38.46) | (11.11, 11.11) | (33.33, 50.00) | (20.83, 20.83) | (54.35, 54.35) | (33.38, 37.36) |
| LLaVA-OV-72B (Li et al., 2024a) | (51.43, 60.00) | (16.67, 41.67) | **(33.33, 33.33)** | (11.54, 19.23) | (90.48, 95.24) | (69.23, 73.08) | (38.89, 38.89) | (66.67, 83.33) | (52.08, 56.25) | (78.26, 78.26) | (50.86, 57.93) |
| Qwen2.5-VL-72B (Bai et al., 2025) | (41.43, 47.14) | **(50.00, 50.00)** | ( 0.00, 16.67) | ( 7.69, 11.54) | (88.10, 92.86) | **(88.46, 96.15)** | **(50.00, 50.00)** | (50.00, 75.00) | (66.67, 68.75) | (78.26, 78.26) | (52.06, 58.64) |
| InternVL3-78B (Zhu et al., 2025) | **(52.86, 65.71)** | (25.00, 33.33) | (20.83, 33.33) | **(30.77, 50.00)** | (90.48, 95.24) | (80.77, 80.77) | (22.22, 22.22) | **(66.67, 83.33)** | **(75.00, 79.17)** | **(84.78, 84.78)** | **(54.94, 62.79)** |
| *Open-source VLMs : > 200B* | | | | | | | | | | | |
| Qwen3-VL-235B (Yang et al., 2025) | **(44.29, 55.71)** | (25.00, 33.33) | (25.00, 41.67) | (23.08, 30.77) | **(100.00, 100.00)** | (88.46, 92.31) | (50.00, 50.00) | (75.00, 83.33) | (64.58, 68.75) | (84.78, 84.78) | (58.02, 64.07) |
| *Proprietary VLMs* | | | | | | | | | | | |
| GPT-5.2 (Singh et al., 2025) | (44.29, 71.43) | (33.33, 41.67) | (16.67, 25.00) | ( 7.69, 7.69) | **(100.00, 100.00)** | (84.62, 96.15) | (55.56, 55.56) | (58.33, 83.33) | (79.17, 79.17) | (82.61, 82.61) | (56.23, 64.26) |
| Claude-Sonnet-4.5 (Anthropic, 2025) | (70.00, 72.86) | (41.67, 41.67) | (54.17, 58.33) | ( 7.69, 15.38) | (90.48, 95.24) | (73.08, 76.92) | (44.44, 44.44) | (75.00, 100.00) | (83.33, 83.33) | (86.96, 86.96) | (62.68, 67.51) |
| Claude-Sonnet-4.5-Thinking (Anthropic, 2025) | (70.00, 78.57) | (25.00, 33.33) | (25.00, 33.33) | (30.77, 34.62) | (90.48, 95.24) | (73.08, 73.08) | (33.33, 33.33) | (83.33, 100.00) | (83.33, 83.33) | (91.30, 91.30) | (60.56, 65.61) |
| Gemini-3-Flash (Google DeepMind, 2025) | (70.00, 85.71) | (75.00, 75.00) | (50.00, 75.00) | (46.15, 65.38) | (100.00, 100.00) | (73.08, 80.77) | (66.67, 66.67) | (83.33, 100.00) | (83.33, 83.33) | (84.78, 84.78) | (73.23, 81.67) |
| Gemini-3-Pro (Google DeepMind, 2025) | (81.43, 81.43) | (58.33, 75.00) | (45.83, 54.17) | (23.08, 38.46) | (100.00, 100.00) | (88.46, 92.31) | (66.67, 66.67) | (100.00, 100.00) | (81.25, 81.25) | (91.30, 91.30) | (73.64, 78.06) |
| *Human reference* | | | | | | | | | | | |
| Human | (77.14, 82.86) | (66.67, 66.67) | (75.00, 83.33) | (76.92, 84.62) | (100.00, 100.00) | (92.31, 100.00) | (66.67, 66.67) | (83.33, 83.33) | (83.33, 83.33) | (91.30, 91.30) | (81.27, 84.21) |

> **Q2. Physical Logic**
> Probes the synthesis of visual perception with fundamental physical laws. The model must actively extract variables and apply principles such as Ohm's Law or rigid body dynamics.
>
> **Standard Prompt:** `"How many grams does the bread in the picture weigh in total?"`
>
> **CoT Prompt:** `"How many grams does the bread in the picture weigh in total?  Please think step by step."`

> **Q3. Contextual Inference**
> Targets visual commonsense, requiring the identification of geolocation, institutional identity, or cultural context from fine-grained visual markers rather than salient foreground objects.
>
> **Standard Prompt:** `"What is the dialect of the region depicted in the picture?"`
>
> **CoT Prompt:** `"What is the dialect of the region depicted in the picture?  Please think step by step."`

> **Q4. Spatial Geometry**
> Addresses the agency in spatial understanding, requiring active identification of reference objects and precise metric estimation (distance, volume) of targets.
>
> **Standard Prompt:** `"What is the capacity (ml) of the bottle in the picture?"`
>
> **CoT Prompt:** `"What is the capacity (ml) of the bottle in the picture?  Please think step by step."`

*Table 5.* **Hierarchical Taxonomy and Statistics of the V-IRD Benchmark.** The benchmark covers four primary domains and 10 fine-grained sub-tasks. For each sub-task, the sample count (#), a precise definition, and a representative query are reported to illustrate the diversity of physical and spatial reasoning challenges.

| Sub-task | # | Description | Sample Questions |
|---|---|---|---|
| *Domain I: Spatial Geometry* | | | |
| Length | 35 | Measure the linear dimension of a target object relative to visual reference scales. | What is the length (cm) of the pencil on the desk? |
| Distance | 13 | Quantify the spatial interval between two distinct entities in 3D space. | What is the distance (cm) between the ID card and the passport? |
| Volume | 12 | Estimate the fluid capacity or displacement volume of containers based on geometry. | What is the capacity (ml) of the transparent bottle? |
| Area | 6 | Calculate the 2D surface coverage of specific planar regions or screens. | What is the area ($cm^2$) of the computer monitor? |
| *Domain II: Contextual Inference* | | | |
| Remark | 24 | Identify specific entity metadata, such as airline brands, logos, or institutional names. | Which country's airline is the person in the picture taking? |
| Environment | 23 | Infer geolocation, temporal context, or cultural dialects from environmental markers. | What is the characteristic dialect of the region depicted? |
| *Domain III: Physical Properties* | | | |
| Temperature | 21 | Deduce thermal states from phase-change artifacts (e.g., steam, ice, condensation). | What is the temperature ($^\circ$C) of the alcohol in the beaker? |
| Weight | 13 | Estimate object mass by integrating visual material properties and approximate volume. | How many grams does the bread weigh in total? |
| *Domain IV: Physical Logic* | | | |
| Electricity | 9 | Apply abstract circuit theories (e.g., Ohm's Law) to visual component states. | What is the resistance ($\Omega$) of the light bulb filament? |
| Kinematics | 6 | Analyze force equilibrium, torque requirements, or motion trajectories. | What is the minimum force (N) required to balance the lever? |

## C. Task Definitions and Dataset Statistics

To rigorously evaluate the physical and spatial agency of Multimodal Large Language Models (MLLMs), we introduce V-IRD, a benchmark grounded in a hierarchical taxonomy. As illustrated in Figure 3, the dataset is stratified into four primary reasoning domains comprising 10 fine-grained sub-tasks. Table 5 provides a comprehensive breakdown of the statistics, definitions, and representative queries for each category.

**Hierarchical Reasoning Domains**

We categorize the reasoning challenges into the following four quadrants, designed to probe distinct facets of visual intelligence:

**1. Spatial Geometry.** This domain addresses the "Agency Deficit" in metric spatial understanding. Unlike generic object detection, tasks in this category (including *Length, Distance, Volume,* and *Area*) require the model to establish an internal metric scale from visual references. As shown in Table 5, this is the largest category in our benchmark, assessing the fundamental capability of implicit information seeking and precise metric estimation from 2D pixel inputs.

**2. Contextual Inference.** Beyond salient foreground objects, this domain targets "Visual Commonsense." It requires the active discovery and identification of subtle cues, which include institutional logos (*Remark*) or environmental markers such as vegetation and architecture (*Environment*), to deduce geolocation, temporal context, or cultural identity.

**3. Physical Properties.** This domain evaluates the capacity to actively deduce invisible object states via secondary visual artifacts. For instance, the *Temperature* task requires inferring thermal states from steam or condensation, while the *Weight* task demands the synthesis of estimated volume and material density.

**4. Physical Logic.** Representing the high level of abstract reasoning, this domain probes the synthesis of visual perception with fundamental physical laws. Models must actively extract implicit visual variables and apply principles such as Ohm's Law (*Electricity*) or rigid body dynamics (*Kinematics*) to solve complex reasoning problems.

## D. Human Evaluation

To assess the degree of alignment between VLMs and human physical understanding, and to establish a robust high-quality reference for the V-IRD benchmark, a human performance evaluation was conducted.

We recruited **3 human participants**, all holding bachelor's degrees with engineering or science backgrounds, representing competent human cognition and reasoning ability in physical and spatial tasks. The assessment process did not set strict time limits, and the average completion time was approximately 5 hours. To guarantee statistical reliability, the participants worked collaboratively to complete **two full iterations** of the entire V-IRD benchmark. Specifically, the workload was distributed among the evaluators such that the dataset was fully annotated two times, providing a consistent consensus metric for human accuracy.

Importantly, the images and text prompts shown to the evaluators were strictly consistent with the inputs provided to the models. This setting aimed to place humans and the models at comparable initial conditions at the input stage, thereby minimizing potential information leakage. We then aggregated these consensus responses to derive the final human performance scores reported in Table 2 and Table 4.

