# OpenReview forum: "Position: The Systemic Lack of Agency in Visual Reasoning"
_ICML.cc/2026/Position_Paper_Track — ICML 2026 Position Paper Track regular_

### Official Review · Reviewer_zd2F · 2026-03-04

**Significance:** 3
**Argument Clarity:** 3
**Rating:** 5
**Confidence:** 4

**Questions:**

Please refer the Strengths And Weaknesses section and address my concerns.

**Alternative Views Section:**

Yes

**Compliance With Llm Reviewing Policy A Conservative:**

Affirmed.

**Discussion Potential:**

3

**Final Justification:**

My concerns have been fully addressed by the author. I maintain my original rating, recommend accepting this paper.

**Paper Summary:**

This paper proposes the position that current Vision-Language Models lack visual agency. Specifically, they can recognize what they are told to look at, but rarely search for unmentioned visual evidence on their own. As a result, these models struggle with implicit reasoning that requires discovering hidden cues in an image. This paper proposes the V-IRD benchmark to expose this gap, showing that strong semantic retrieval does not translate into active, self-directed reasoning.

**Position:**

Yes

**Position In Title:**

Yes

**Related Work:**

3

**Strengths And Weaknesses:**

This paper introduces a position that is popular to discuss and provides strong data, examples, and analysis to support it. The entire paper is well-motivated, readable, and has a good writing structure. However, I still have the following concerns, which are the weaknesses of this paper:

1. The V-IRD appears to be relatively small. Have its inclusion and bias been screened or professionally addressed by experts? This is crucial for future discussion and analysis.
2. The human evaluation criteria could need more detail. Currently, it's unclear whether there are leakage issues, such as whether humans know to look for reference points.
3. I believe the authors' overall assumptions may be somewhat biased (see Line 79, "By enforcing a strict information gap via Target-Exclusive Prompting"). This seems limiting. Could the discussed issue be addressed through better prompt engineering or chain-of-thought?
4. The paper contains some minor errors.

**Support:**

4

---

> ### Author Rebuttal · Authors · 2026-03-31
>
> **Weakness 1: dataset scale**
>
> The construction of V-IRD focuses on challenging implicit reasoning and is accompanied by careful quality screening. The dataset was subjected to strict manual screening and screening to better ensure the rigor of the samples in terms of physical laws and geometric scales. Regarding bias control and inclusiveness, since the core task in V-IRD is grounded mainly in reasoning via objective physical laws, it may be less directly influenced by certain social biases than benchmarks centered on human attributes or social semantics. In addition, to reduce the distribution bias of the model towards a single image style, we adopted a multi-source acquisition strategy covering real photos, network collection, and preliminary filtering of AI-generated images, aiming to improve the diversity and inclusiveness of visual scenes under the current data scale.
>
> **Weakness 2: human evaluation details**
>
> To strengthen the rigor of the assessment, all three assessors held bachelor's degrees. Their performance can be viewed as representing reasonably strong human cognition and reasoning ability. The assessment process did not set strict time limits, and the average completion time was approximately 5 hours. Importantly, the images and text prompts shown to the assessors were consistent with the inputs that were provided to the model. This setting aimed to place humans and the model at relatively similar initial conditions at the input stage, thus reducing concerns about potential information leakage arising from the experimental design. We will add these implementation details more explicitly in the revised manuscript.
>
> **Weakness 3: Overall assumptions may be biased**
>
> Target-Exclusive Prompting should not necessarily be interpreted as a biased restriction. Instead, it more closely resembles the natural way questions are asked in real-world settings. When humans are faced with exactly the same input instructions, they are often able to understand the questions smoothly and spontaneously search for implicit references, demonstrating corresponding visual initiative. If the prompt contains more key information, it can potentially improve the model to a certain extent.
>
> **Weakness 4: minor mistakes**
>
> Thank you for the reminder. We will conduct a comprehensive verification and correction of the entire text.

---

> > ### Author Rebuttal · Reviewer_zd2F · 2026-04-01
> >
> > Thank you to the author for the detailed rebuttal. My concerns regarding the data,evaluation details, and its assumptions have been fully addressed by the author. Therefore, I maintain my original rating, recommend accepting this paper at this stage.

---

### Official Review · Reviewer_Qma6 · 2026-03-11

**Significance:** 3
**Argument Clarity:** 3
**Rating:** 5
**Confidence:** 3

**Questions:**

1. Given the small sample sizes in several V-IRD subtasks, how can the authors demonstrate that the observed performance degradation on such limited data is statistically significant?

2. The paper attributes the models' failures to a systemic "Lack of Agency". However, could this merely be an artifact of overfitting to highly explicit instruction data during the fine-tuning phase?

3. In the information injection experiments, performance improves substantially as prompts become more explicit. Is this improvement simply because the refined prompts better align with the explicit question-answering formats seen during pre-training?

**Alternative Views Section:**

Yes

**Compliance With Llm Reviewing Policy A Conservative:**

Affirmed.

**Discussion Potential:**

3

**Final Justification:**

The author's response reinforces my initial positive evaluation. I elect to maintain my current rating.

**Paper Summary:**

This position paper argues that current Vision-Language Models  suffer from a systemic lack of Agency in visual reasoning, relying heavily on instruction-driven passive capacity rather than possessing the visual agency needed to autonomously identify unprompted visual evidence. To quantify this deficit, the authors introduce the V-IRD benchmark, utilizing a goal-exclusive prompting strategy that obscures key reference clues to force autonomous information retrieval. Extensive experiments reveal that despite having adequate foundational recognition and parametric knowledge, current models experience severe performance degradation when required to actively discover implicit cues. These findings demonstrate that simply scaling model size does not organically yield true visual agency, highlighting an urgent need for the multimodal research community to shift from passive semantic matching toward developing architectures inherently capable of active exploration and visual discovery.

**Position:**

Yes

**Position In Title:**

Yes

**Related Work:**

3

**Strengths And Weaknesses:**

Strengths

1. The clear distinction between visual capacity and visual agency successfully exposes a critical blind spot in current evaluation frameworks, demonstrating that most existing VLM benchmarks merely test guided recognition rather than autonomous reasoning.

2. By sequentially verifying foundational knowledge, testing under implicit conditions, and demonstrating performance recovery through explicit information injection, the authors convincingly prove that model failures stem from a lack of active visual exploration rather than computational inability.

Weaknesses

1. Despite strict manual curation, the V-IRD benchmark's small scale raises concerns regarding statistical significance.The results are highly vulnerable to statistical variance under current LLM evaluation standards.

2. While this position paper successfully diagnoses the problem, its proposed remedies remain superficial. The authors conclude by calling for training objectives that "promote autonomous visual discovery and active perception," but they do not provide any concrete architectural designs or algorithmic blueprints.

**Support:**

3

---

> ### Author Rebuttal · Authors · 2026-03-31
>
> **Weakness 1 and Question 1: dataset scale**
>
> The construction strategy of V-IRD mainly focuses on challenging implicit information reasoning. To ensure the physical validity of the reasoning process to a large extent, we conducted meticulous manual screening of the data and restricted the information accessible to the model through the Target-Exclusive Prompting strategy. Preliminary observations based on the threshold-based accuracy show that even when the error tolerance is relaxed to 20%, multiple models still exhibit extremely low accuracy in complex tasks. This performance decline seems to go beyond the influence of random variation alone. This might indicate that the current scale of labeled samples helps mitigate the influence of statistical variance and reveals the problem of the model's lack of visual agency across architectures.
>
> **Weakness 2: Regarding the solution**
>
> Thank you to the reviewers for your insightful feedback. As a position paper, the primary goal of this work is to highlight the blind spots in current evaluations of model capabilities, and to question the relatively optimistic assumption that "only by expanding the parameter size and fine-tuning with high-quality instructions can deep visual reasoning naturally emerge". Although this article does not yet present a complete architecture blueprint or specific algorithm design, from a conclusion-oriented viewpoint, we hope to establish preliminary quantitative benchmarks and problem diagnoses through which we can provide some exploratory directions for the future algorithm evolution and optimization routes in the multimodal field.
>
> **Question 2: artifact of overfitting**
>
> We agree that the observed failures might partially be an artifact of overfitting to explicit instruction data. However, this overfitting tends to act as a core driver of the systemic lack of visual agency itself. Since current training paradigms frequently present queries with explicit key information, models are often induced to overly rely on textual priors rather than actively discovering implicit visual cues.
>
> **Question 3: information injection**
>
> Regarding the clear performance gains observed in medium-sized models, this improvement does not seem to result solely from a simple alignment between the prompt format and the pre-training question-answering paradigm. Instead, it may indicate that guidance from explicit external physical cues can more effectively compensate for the lack of visual agency in such models.

---

> > ### Author Rebuttal · Reviewer_Qma6 · 2026-04-03
> >
> > Thank you for the response. I will maintain my positive score.

---

### Official Review · Reviewer_68C5 · 2026-03-12

**Significance:** 3
**Argument Clarity:** 4
**Rating:** 5
**Confidence:** 3

**Questions:**

The questions are summarized in the weaknesses, which need further explanation.

**Alternative Views Section:**

Yes

**Compliance With Llm Reviewing Policy A Conservative:**

Affirmed.

**Discussion Potential:**

3

**Final Justification:**

My concerns are addressed. I will keep my rating.

**Paper Summary:**

This paper advocates the position that current Vision-Language Models suffer from a systemic lack of visual agency, limiting their ability to perform implicit visual reasoning in unconstrained environments. The authors discuss an important concept by formally distinguishing between visual capacity, which is the ability to perceive details when explicitly guided, and visual agency, which is the autonomous intent to search for unmentioned visual evidence to construct physical arguments. To empirically validate this deficit, the authors introduce the Visual Implicit Reasoning Benchmark, or V-IRD, that strictly enforces an information gap using target-exclusive prompting across domains like spatial geometry and physical logic. Through extensive evaluations and diagnostic experiments, the study reveals that the primary bottleneck for mainstream models is an active discovery failure rather than a lack of fundamental parametric knowledge or logical calculation capability. Overall, a critical question presented by the study is whether contemporary vision-language models function as grounded visual reasoners or merely as passive probabilistic semantic retrievers, ultimately concluding that future progress necessitates training objectives that explicitly promote active perception rather than relying solely on model scaling or explicit prompting.

**Position:**

Yes

**Position In Title:**

Yes

**Related Work:**

3

**Strengths And Weaknesses:**

### Strengths
1. The paper provides a compelling and highly relevant critique of current multimodal evaluation paradigms. The authors discuss an important concept by formally separating visual capacity from visual agency, arguing effectively that current models act as passive readers rather than active explorers.
2. A major strength is the experimental design, particularly the prerequisite validation phase. This successfully isolates the root cause of reasoning failures by proving that models possess the necessary parametric knowledge but simply fail to deploy it autonomously.
3. The diagnostic error decomposition into discovery, association, and logical calculation stages provides a highly granular and insightful understanding of exactly where cognitive breakdowns occur in large vision-language models.

### Weaknesses
1. The scale of the V-IRD benchmark appears relatively modest. The fine-grained sub-tasks contain a limited number of samples each, which might benefit from a larger and more diverse corpus to ensure greater statistical robustness across different model architectures.
2. The evaluation is entirely restricted to static 2D images.
3. The human baseline is established using only three participants, which may not provide a definitive or statistically robust upper-bound for human performance.

**Support:**

3

---

> ### Author Rebuttal · Authors · 2026-03-31
>
> **Weakness 1: dataset scale**
>
> The construction of V-IRD targets highly challenging implicit information reasoning. To preserve the physical validity of the reasoning process, we applied rigorous manual screening of the data, and also restricted model access to target-related information through the Target-Exclusive Prompting strategy. The current sample size is often sufficient to reveal the systematic deficiencies of the model across architectures, demonstrating the corresponding statistical robustness. Under threshold-based accuracy, some challenging tasks remain at extremely low accuracy even with a 20% error margin, and this decline is unlikely to be attributable to random variation alone. This indicates that, to some extent, current labeled samples can reveal the systematic problem of visual proxy deficiency across architectures, possessing preliminary statistical significance in reflecting this phenomenon.
>
> **Weakness 2: 2D picture**
>
> Perception in the real world often relies on videos or 3D environments, whereas visual reasoning is often based on static 2D images. Our experiments have shown that if the model fails to establish an effective association between text and image, even in fully informative 2D images, such errors will only worsen in the exploration of more complex 3D spaces and video. Additionally, we conducted preliminary video reasoning experiments. For the objective results, please refer to Table 3 at https://anonymous.4open.science/r/rebuttal-3F78. The results indicate that errors still persist when models process high-dimensional temporal videos. This phenomenon seems to support the perspective that merely introducing temporal dimensions may not naturally resolve the lack of visual agency.
>
> **Weakness 3: Human Evaluation**
>
> Introducing the human benchmark provides an intuitive reference for the "objective solvability" of the task. All three test participants held bachelor's degrees and can be regarded as representing competent human reasoning ability. To ensure assessment quality, each participant spent approximately 5 hours completing the questions independently and under controlled conditions, which helped minimize the risk of information leakage. This carefully designed process helps establish a high-quality reference for task performance and, to a considerable extent, demonstrates the practical solvability of this type of problem. Since the term "upper limit" may be inappropriate in this context and could easily cause misunderstanding, we will revise the manuscript accordingly and replace it with the more objective expression "high-quality reference".

---

> > ### Author Rebuttal · Reviewer_68C5 · 2026-04-04
> >
> > I thank the rebuttal of the authors. My concerns are addressed. I will keep my rating.

---

### Official Review · Reviewer_4a33 · 2026-03-12

**Significance:** 3
**Argument Clarity:** 3
**Rating:** 4
**Confidence:** 4

**Questions:**

Please see some of the questions already mentioned above.
Additionally:
- Will the benchmark be fully released?
- When a model solves a task correctly, do they show visual agency? In other words, do they get the answer right for the correct reasons? How does the benchmark mitigate shortcuts?
- How do current frontier models acquire the capabilities to solve a reasoning chunk of the benchmark (larger models)? Are they likely implicitly induced or learned through supervision? A discussion on this could be very interesting if the position implies that current training pipelines do not cover agency in visual reasoning.

Suggestion:
- Fig. 3 is confusing at first. I suggest to put the super-category on the outside and merge the percentage size with the sub-category to make it clear they belong together. I would also not shrink the pie chart pieces as I am not sure what the purpose is here.

**Alternative Views Section:**

Yes

**Compliance With Llm Reviewing Policy A Conservative:**

Affirmed.

**Discussion Potential:**

2

**Final Justification:**

The rebuttal partially addressed my concerns.

I believe the results in Tab. 3 and the discussion around it is misleading by itself, even through the overall conclusion are not necessarily wrong taking the whole paper into account. I would find a less biased experimental design more appropriate.

While the authors have improved the argumentation around Fig. 5 by being mostly objective about the results, the results remain unintuitive which weakens the position slightly.

For these reasons I maintain my original score, but still think that the paper is above the acceptance threshold.

**Paper Summary:**

The paper argues that current vision-language models (VLMs) fundamentally lack agency when performing visual reasoning tasks. The presented position states that, when VLMs perform well on that that explicitly present the target objects and/or the required steps for reasoning relevant to solve the visual reasoning task. In situations where supporting visual evidence is required but not mentioned by the task prompt, VLMs fail as they do not autonomously explore and seek out additional information that may be in the background or separate from the target object. The paper presents a new benchmark, V-IRD, to probe for visual implicit reasoning in VLMs and evaluates open-source and proprietary models of various sizes.

**Position:**

Yes

**Position In Title:**

Yes

**Related Work:**

2

**Strengths And Weaknesses:**

Strengths:
- The position is well motivated, presented, and clearly separated from/put in relation to adjacent topics such as hallucinations and physical reasoning.
- The proposed benchmark provides sufficient evidence for the paper's position. The paper extensively evaluates VLMs across a wide range of sizes.
- An extended analysis attributes the failures of current VLMs to an agency deficiency. These results are mostly convincing.
- The paper addresses the obvious scaling position that opposes this view.
- The position is likely to resonate with most of the ICML community and may spark some discussion with people believing in the scaling hypothesis.

Weaknesses:
- The main concern resolves around the benchmark and experiments that are used as evidence for the position:
  - A key premise for the benchmark is that agency is required to solve the tasks. However, there is no strong evidence that this benchmark cannot be solved without agency. The trend of larger models performing better could either be attributed to scale, or to included agentic tasks during training. One option to validate that the tasks truly measures implicit reasoning could have been to try fine-tuning a smaller model on task examples without providing explicit supervision beyond question-answer pairs. If the benchmark cannot be "cheated" without agentic reasoning, the model should not perform better after fine-tuning.
  - Crucial details about the benchmark are missing such as: How many samples does the benchmark have? What model was use for AI generated images? What is the proportion of images from manual photography, web crawling, and AI generation? How were the prompts selected?
  - The reasoning behind using threshold accuracy is not clear. Why not just report an average, normalized, continuous score of relative error?
  - The results on agency vs capacity deficit (Tab. 3) are skewed towards agency. As agency is evaluated first, it gets credit for when a model might fail at both. In that case, capacity failure is not attributed, however. If it was done the other way around, the conclusions, e.g., those on Gemini could be different. While everything in general points to agency failures, it remains a flawed metric/evaluation.
  - The results from Fig. 5 do not seem to line up with the discussion in section 4.6. Only InternVL3.5-38B and Qwen3-VL-8B seem to have a clear improvement in accuracy going from 0 to 3. Most other models even decrease in performance which is counter-intuitive.
  - There is no discussion of the human study and its results. There is some additional information in the supplementary, but it presents the human results as "upper-bound". At the same time, experts tried to solve these tasks. This raises some concerns about where the gap to 100% accuracy comes from. How was the benchmark validated if it cannot be solved by humans? If models like Gemini are already close to the upper-bound, where does the gap between human and AI lie?

- Discussion on existing solutions: It would have helped to put existing solutions into perspective and argue whether or not they follow the paper's position. For instance, do reasoning approaches that use RL to let the model learn to solve problems agree with this position and already present a reasonable solution?

- The position is not a particularly novel take. While I believe that it is valuable to further highlight this deficiency of current VLMs, work on reasoning is actively exploring solutions to these problems.

**Support:**

3

---

> ### Author Rebuttal · Authors · 2026-03-31
>
> **Weakness 1: Regarding the benchmark**
>
> Current observations indicate that scaling up model size often improves performance, likely because large models encounter similar correlation patterns during pre-training. However, static training data can rarely capture the full complexity of the physical world. We followed your exact proposal: we fine-tuned Qwen2.5-VL-3B, Qwen3-VL-2B/4B, InternVL3.5-2B/8B on 91 question-answer pairs, and tested on the remaining samples. In the anonymous link https://anonymous.4open.science/r/rebuttal-3F78 Table 1, the results suggest that fine-tuning doesn't improve the implicit reasoning abilities of these models, but reduced it, firmly validating your hypothesis. This indicates that the tasks truly measure implicit reasoning, and our benchmark can't be manipulated to achieve a favorable result.
>
> **Weakness 2: Regarding the benchmark details**
>
> V-IRD contains 169 samples: hand-photographed (60/35.5%), web-crawled (64/37.9%), and AI-generated images (45/26.6%) by Nano Banana. Prompts were manually drafted as one or two sentences describing only the target subject, without any background or contextual details.
>
> **Weakness 3: Regarding the reasoning for using threshold accuracy**
>
> The average or normalized continuous relative error score is misleading, as hallucinations introduce outliers that distort the aggregate evaluation. Such metrics overweight rough guesses. Threshold-based accuracy can effectively eliminate vague guesses, supported by previous work  [1].
> [1] Thinking in space: How multimodal large language models see, remember, and recall spaces. CVPR 2025.
>
> **Weakness 4: The results on agency vs capacity deficit**
>
> Implicit visual reasoning follows a coherent, unidirectional logical chain: actively discover implicit information -> associate with the query target -> accurately calculate attribute values. Each step can proceed only after the previous one succeeds.
>
> **Weakness 5: Figure 5 results**
>
> Figure 5 shows clear scale-dependent effects of information injection. Larger-scale models (Gemini3-pro, InternVL3-78B) show no expected gains, and some even decline, possibly because the extensive pre-trained knowledge conflicts with injected physical information. Medium-sized models (InternVL3-14B, Qwen3-VL-8B) improve noticeably, indicating the benefit of explicit guidance. Small 2B-scale models (InternVL3-2B) shows little change, likely due to limited capacity.
>
> **Weakness 6: human study and its results**
>
> Human performance serves as a high-quality reference for measuring agency. Human errors arise mainly from visual estimation, not a lack of visual agency. Unlike current AI, humans actively seek implicit clues under insufficient information. To avoid misunderstanding, we replace "upper limit" with "high-quality reference benchmark" in the revised manuscript.
>
> **Weakness 7: Discussion on existing solutions**
>
> Due to visual-textual discrepancies, mainstream RL designs favor text-oriented logic and struggle to facilitate the active search for latent visual cues. Thus, they find it challenging to address the lack of visual agency required for visually implicit reasoning.
>
> **Weakness 8: The novelty of the position**
>
> Though reasoning research is advancing rapidly, the active acquisition of visual cues remains overlooked. Existing paradigms mainly optimize the internal logical paths for the given input information (whether it is text or passively received visual data), while our position's value lies in calling for an expansion of the focus to the active acquisition of visual information. We hope this perspective can inspire future multimodal reasoning research.
>
> **Question 1: benchmark release**
>
> This benchmark will be fully released.
>
> **Question 2: visual agency**
>
> We analyzed the Chain of thought trajectories of the successful cases. Please refer to Table 2 at https://anonymous.4open.science/r/rebuttal-3F78, most correct predictions rely on explicit recognition and logical association of implicit reference objects. The Geometric is a notable exception, where performance remains relatively low but shows a scaling trend. To mitigate prior knowledge shortcuts and filter out vague guesses, the benchmark combines Target-Exclusive Prompting with the precision-sensitive threshold accuracy.
>
> **Question 3: Frontier Models**
>
> Frontier models likely develop this reasoning via massive data rather than explicit visual agency supervision. While some models indicate signs of visual initiative, the precise mechanisms often remain difficult to fully explain. Although current training pipelines appear to have limited coverage of active exploration, our benchmark may serve as a quantifiable starting point, potentially allowing us to analyze the origins of this ability across different architectures in a more detailed manner.
>
> **Suggestion: Regarding Figure 3**
>
> Thanks for the suggestion. We have updated Figure 3, provided as Figure 1 at  https://anonymous.4open.science/r/rebuttal-3F78.

---

> > ### Author Rebuttal · Reviewer_4a33 · 2026-04-02
> >
> > I thank the authors for their clarifications that partially addressed my concerns.
> >
> > I still disagree with W4. While I acknowledge the sequential processing, it does not change the fact that it leads to a bias towards attributing a model's deficit to these capabilities. Either the framing needs to be updated to simply state _when_ the model fails, or the capacity deficit needs to be evaluated with an oracle agency step. This experiment feels too much like pushing the narrative with a flawed evaluation/conclusion.
> >
> > While the clarifications on W5 objectively states some observations, it is not clear why these are happening as they go against intuition. The manuscript text is imprecise and it is not clear how this is fixed.
> >
> > It remains unclear how to interpret the human results (W6). If human experts cannot solve these tasks, why should be expect AI to do so? Maybe there is a good reason, but it is not clear whether the benchmark/metric is flawed or whether some examples are even too hard for humans. If visual estimation is an issue, how can the GT answer be validated, especially for web-crawled and AI images?
> >
> > I think there are sufficient works in this direction, including reasoning models using RL [1,2], that could warrant a discussion. There are just two recent works in this area that has seen increasing development especially since 2025.
> >
> > For these reasons I maintain my score.
> >
> > [1] Lin et al., AdaptVision: Efficient Vision-Language Models via Adaptive Visual Acquisition, CVPR 2026
> > [2] Zheng et al., DeepEyes: Incentivizing "Thinking with Images" via Reinforcement Learning, ICLR 2026

---

> > > ### Author Response · Authors · 2026-04-05
> > >
> > > **Reply to Question 1**
> > >
> > > &emsp;Thanks for the feedback. **In our quantitative analysis of failure stages**, we evaluated model performance across three sequential reasoning steps: active discovery of implicit information, association of the reference with the target, and execution of the final calculation. The results show the overall proportion of errors occurring at the initial step. Despite achieving the highest overall accuracy, about 60% of Gemini's failures originate from this first stage. **Furthermore, our Explicit Information Injection experiment** shows that for medium-scale models, once the visual cues are provided, their capabilities in this aspect exhibit a upward trend. These analysis reveals that the initial stage of actively discovering implicit information accounts for the majority of errors.This analysis indicates that the initial phase of actively discovering implicit information is a primary contributor to the overall error rate. Therefore, the primary bottleneck leading to failure is the systemic lack of active visual agency at the initial step.
> > >
> > > **Reply to Question 2**
> > >
> > > &emsp;Thanks for the reminder. We will update the manuscript to provide a more comprehensive explanation of the scale-dependent phenomena observed in Figure 5. Specifically, we have replaced the "Results and Implications" part in Section 4.6 of the original manuscript with the following discussion:
> > >
> > > &emsp;For large-scale models, the highly explicit physical information injected into the prompts may conflict with their extensive pre-trained knowledge. Such models generally depend on robust parametric priors. Therefore, discrepancies between the provided physical cues and their inherent knowledge may induce reasoning conflicts, leading to stagnant or diminished performance. Conversely, medium-scale models demonstrate an improvement. This positive trend indicates that explicit external guidance mitigates their intrinsic deficit in active visual exploration to a certain degree. Meanwhile, the performance of small-scale models remains relatively static. This limitation is presumably attributable to their restricted baseline capacity, which impedes their ability to process and integrate complex visual cues into the final reasoning procedures.
> > >
> > > **Reply to Question 3**
> > >
> > > &emsp;We thank the reviewer for this insightful question. Sub-optimal human performance highlights a cognitive divergence rather than a flawed benchmark. Humans inherently possess the Visual Agency to actively locate implicit references, yet struggle with precise numerical estimation. Conversely, VLMs excel in computational accuracy but lack the subjective initiative to seek out those references. Our benchmark challenges VLMs to bridge this exact gap.
> > >
> > > &emsp;Regarding Ground Truth validation, we explicitly avoid ambiguous human visual estimation. Our annotations strictly rely on verifiable, objective facts across all data sources:
> > >
> > > 1. **Manual Photography**: GT is established via direct physical measurement of the target objects prior to image capture.
> > >
> > > 2. **Web-Crawled Imagery**: GT is extracted from objective source metadata and independently cross-validated.
> > >
> > > 3. **AI-Generated Imagery**: We strictly restrict AI-generated images to tasks where the GT can be derived purely through objective logical deduction. For example, an object's mass GT is logically deduced by reading the exact values of generated weights on a balance scale. All synthetic images are also rigorously screened for physical consistency.
> > >
> > > **Reply to Question 4**
> > >
> > > &emsp;We sincerely thank the reviewer for highlighting these relevant, concurrent works. As they were released immediately before or after the submission deadline, we will gladly situate our research alongside them in the revised Related Work section.
> > >
> > > &emsp;Conceptually, our work and the suggested papers address complementary dimensions of visual reasoning. AdaptVision [1] and DeepEyes [2] elegantly employ reinforcement learning to enhance the execution of perception by actively zooming in on local details. In contrast, our paper formalizes a related foundational concept: the Visual Agency deficit.
> > >
> > > &emsp;The effective guidance of such zooming mechanisms is tied to a model's underlying capacity to autonomously seek out unmentioned visual clues. As demonstrated by our V-IRD benchmark, if a model lacks the intrinsic agency to initiate autonomous information retrieval, it may struggle to determine where or why to focus its attention. Therefore, we view our work as complementary: by exploring this underlying agency bottleneck, our findings provide a perspective that could further inspire and enhance such active visual reasoning systems.
> > >
> > > **We sincerely hope these responses resolve any concerns. We welcome any further questions and would greatly appreciate your feedback.**

---

### Decision · Program_Chairs · 2026-04-30

**Decision:**

Accept (regular)

**Comment:**

This paper convincingly argues that current Vision-Language Models lack "visual agency," operating as passive semantic retrievers rather than active visual explorers. To evaluate this limitation, the authors introduce the V-IRD benchmark, revealing that models struggle to autonomously discover unprompted visual cues required for implicit reasoning.
Reviewers 68C5 and Qma6 praised the paper's clear distinction between visual capacity and agency, noting it exposes a critical blind spot in multimodal evaluation. Reviewer 4a33 highlighted the extensive and well-motivated evaluation across various model sizes. The primary weakness noted by all reviewers (4a33, 68C5, Qma6, zd2F) is the modest scale of the V-IRD benchmark, which raises minor statistical robustness concerns. Additionally, Qma6 noted the lack of concrete algorithmic blueprints for future architectures.
Despite these limitations, the paper is highly relevant, well-argued, and offers an essential diagnostic framework for understanding VLM reasoning failures.